# Gut Microbiome and Its Cofactors Are Linked to Lipoprotein Distribution Profiles

**DOI:** 10.3390/microorganisms10112156

**Published:** 2022-10-31

**Authors:** Josué L. Castro-Mejía, Bekzod Khakimov, Violetta Aru, Mads V. Lind, Eva Garne, Petronela Paulová, Elnaz Tavakkoli, Lars H. Hansen, Age K. Smilde, Lars Holm, Søren B. Engelsen, Dennis S. Nielsen

**Affiliations:** 1Department of Food Science, University of Copenhagen, 1958 Frederiksberg C, Denmark; 2Department of Nutrition, Exercise and Sports, University of Copenhagen, 1958 Frederiksberg C, Denmark; 3Department of Biomedical Sciences, University of Copenhagen, 2200 Copenhagen N, Denmark; 4Biomedical Research Center, Institute of Experimental Endocrinology, University Science Park for Biomedicine, Dúbravská cesta 9, 94505 Bratislava, Slovakia; 5Swammerdam Institute for Life Sciences, University of Amsterdam, Postbus 94215, 1090 GE Amsterdam, The Netherlands; 6Department of Plant and Environmental Sciences, University of Copenhagen, 1871 Frederiksberg C, Denmark; 7School of Sport, Exercise and Rehabilitation Sciences, University of Birmingham, Birmingham B15 2TT, UK

**Keywords:** gut microbiota, SCFAs, lipoproteins distribution, HDL, ^1^H-NMR

## Abstract

Increasing evidence indicates that the gut microbiome (GM) plays an important role in dyslipidemia. To date, however, no in-depth characterization of the associations between GM with lipoproteins distributions (LPD) among adult individuals with diverse BMI has been conducted. To determine such associations, we studied blood-plasma LPD, fecal short-chain fatty acids (SCFA) and GM of 262 Danes aged 19–89 years. Stratification of LPD segregated subjects into three clusters displaying recommended levels of lipoproteins and explained by age and body-mass-index. Higher levels of HDL2a and HDL2b were associated with a higher abundance of *Ruminococcaceae* and *Christensenellaceae*. Increasing levels of total cholesterol and LDL-1 and LDL-2 were positively associated with *Lachnospiraceae* and *Coriobacteriaceae*, and negatively with *Bacteroidaceae* and *Bifidobacteriaceae*. Metagenome-sequencing showed a higher abundance of biosynthesis of multiple B-vitamins and SCFA metabolism genes among healthier LPD profiles. Metagenomic-assembled genomes (MAGs) affiliated to *Eggerthellaceae* and *Clostridiales* were contributors of these genes and their relative abundance correlated positively with larger HDL subfractions. The study demonstrates that differences in composition and metabolic traits of the GM are associated with variations in LPD among the recruited subjects. These findings provide evidence for GM considerations in future research aiming to shed light on mechanisms of the GM–dyslipidemia axis.

## 1. Introduction

Cholesterol is essential for keeping cellular integrity and is an important precursor for steroid hormones and bile acids [1]. However, alterations of the cholesterol metabolism and consequent dyslipidemia have been associated with various diseases, including atherosclerosis and cardiovascular diseases (CVD) [2], as well as breast cancer [3].

Recent advances in metabolomics research have allowed large-scale and high-throughput profiling of lipoprotein distribution’s (LPD) in human blood plasma based upon their composition and concentration [4,5,6]. It has been hypothesized that numerous medical conditions such as glucose intolerance, type-2 diabetes, myocardial infarction, ischemic stroke and intracerebral hemorrhage, might be associated with lower blood levels of larger HDL particles (e.g., HDL2a and HDL2b) and a higher content of triglycerides within the lipoprotein particles [7,8].

During the last decade it has been shown that alterations in gut microbiome (GM) composition contribute to the development and progression of several metabolic and immunological complications [9]. Furthermore, a handful of recent studies on different cohorts have also demonstrated that the changes in intestinal microbiota are highly correlated to variations in levels of lipoproteins in blood [10,11,12], as well as to promote atherosclerosis [13], and regulate cholesterol homeostasis [14], and partly mediated by bile acid and short-chain fatty acids (SCFA) signaling [15].

The relationship between GM and LPD has only been scarcely investigated. Recently, Vojinovic et al. [5] reported the association of up to 32 GM members with very-low-density (VLDL) and high-density (HDL) subfractions. Positive correlations between a number of Clostridiales members with large particle size subfractions of HDL were elucidated. In other studies, focusing on total lipoproteins fractions, an increasing abundance of GM members affiliated to the *Erysipelotrichaceae* and *Lachnospiraceae* families have been linked to elevated levels of total cholesterol and low-density lipoproteins (LDL) [10,11,12]. Interestingly, common gut microbes like *Lactobacillaceae* members have been reported to assimilate and lower cholesterol concentrations from growth media and incorporate it into their cellular membrane [16], whereas butyrate-producing *Roseburia intestinalis* has been found to increase fatty acid utilization and reduce atherosclerosis development in a murine model [17].

Thus, with the aim of gaining a deeper understanding of the relationship between GM and LPD in blood, we carried out a detailed compositional analysis of GM, its metabolic functions, and studied its associations with blood lipoproteins quantified using a recently developed method based on proton (^1^H) nuclear magnetic resonance (NMR) spectroscopy [6]. We determined covariations between larger HDL subclasses and lower total cholesterol with a several *Clostridiales* (*Ruminococcaceae* and *Lachnospiraceae*) and *Eggerthelalles* members, whose metabolic potential is linked to biosynthesis of cofactors essential for carrying out lipid metabolism.

## 2. Materials and Methods

### 2.1. Study Participants

Two hundred and sixty-two men and women participants older than 20 years, who had not received antibiotic treatment 3 months prior to the beginning of the study and who had not received pre- or probiotics 1 month prior to the beginning of the study, were included as part of the COUNTERSTRIKE (COUNTERacting Sarcopenia with proTeins and exeRcise–Screening the CALM cohort for lIpoprotein biomarKErs) project (counterstrike.ku.dk). The CALM (Counteracting Age-related Loss of Skeletal Muscle Mass–clinical trials: NCT02115698) cohort provided sample-material at baseline from participants older than 64 years (*n* = 62), while COUNTERSTRIKE recruited participants between 20 to 64 years (*n* = 200). Pregnant and lactating women, as well as participants previously suffering from cardiovascular diseases (CVD), diabetes, or chronic gastrointestinal disorders, were excluded from the study. Elevated plasma lipid levels or cholesterol-lowering drugs was not an exclusion criteria (8 participants from the CALM cohort were under treatment for high blood pressure using statins). All participants were recruited via press and online announcements and gave written consent to participate in the study.

### 2.2. Lipoprotein Distribution Profiles

The human blood plasma lipoproteins were quantified using SigMa LP software [18]. The SigMa LP quantifies lipoproteins from blood plasma or serum using optimized partial least squares (PLS) regression models developed for each lipoprotein variable using one-dimensional (1D) ^1^H NMR spectra of blood plasma or serum and ultracentrifugation based quantified lipoproteins as response variables as determined in Khakimov et al. [6].

### 2.3. Short Chain Fatty Acids (SCFAs) Quantification

Targeted analysis and quantification of SCFA on fecal slurries were carried out as recently described [19].

### 2.4. Samples Processing, Library Preparation and DNA Sequencing

Fecal samples were collected and kept at 4 °C for maximum 48 h after voidance and stored at −60 °C until further use. Extraction of genomic DNA and library preparation for high-throughput sequencing of the V3-region of the 16S rRNA gene was performed as previously described [19]. Shotgun metagenome libraries for sequencing of genome DNA were built using the Nextera XT DNA Library Preparation Kit (Cat. No. FC-131-1096) and sequenced with Illumina HiSeq 4000 by NXT-DX.

### 2.5. Analysis of Sequencing Data

The raw dataset containing pair-end amplicon reads was analyzed following recently described procedures [19]. The metabolic potential of the amplicon sequencing dataset was determined through PICRUSt [20], briefly, zero-radius operational taxonomical units (zOTUs) abundances were first normalized by copy number and then KEGG orthologues was obtained by predicted metagenome function.

For shotgun sequencing, the reads were trimmed from adaptors and barcodes and the high-quality sequences (>99% quality score) using Trimmomatic v0.35 [21] with a minimum size of 50nt were retained. Subsequently, sequences were dereplicated and check for the presence of Phix179 using USEARCH v10 [22], as well as human and plant genomes associated DNA using Kraken2 [23]. High-quality reads were then subjected to within-sample denovo assembly using Spades v3.13.1 [24], and the contigs with a minimum length of 2000 nt were retained. Within-sample binning was performed with metaWRAP [25] using Metabat1 [26], Metabat2 [27] and MaxBin2 [28], and bin-refinement [29] was allowed to a ≤10% contamination and ≥70% completeness. Average nucleotide identity (ANI) of metagenome assembled genomes (MAGs), was calculated with fastANI [30] and distances between MAGs were summarized with bactaxR [31]. To determined abundance across samples, reads were mapped against MAGs with Subread aligner [32] and a contingency-table of reads per Kbp of contig sequence per million reads sample (RPKM) was generated. Taxonomic annotation of MAGs was determined as follows: ORF calling and gene predictions were performed with Prodigal [33], the predicted proteins were blasted (blastp) against NCBI NR bacterial and archaeal protein database. Using Basic Sequence Taxonomy Annotation tool (BASTA) [34], the Lowest Common Ancestor (LCA) for every MAG was estimated based on percentage of hits of LCA of 60, minimum identity of 0.7, minimum alignment of 0.7 and a minimum number of hits for LCA of 10.

To determine the metabolic potential of metagenomes, ORF calling and gene predictions (similar as above) were performed on both, binned and unbinned contigs, and the predicted proteins were subsequently clustered at 90% similarity using USEARCH v10. To assign functions, protein sequences were blasted (90% id and 90% cover query) against the integrated reference catalog of the human gut microbiome (IRCHGM) [35], while using only target sequences containing KEGG ortholog entries. Similar to the above, to determine abundance of protein-encoding genes across metagenomes, reads were mapped against protein clusters (PC) with DIAMOND [36] and a contingency-table of reads mapped to PCs was also generated. To avoid bias due to sequencing depth across protein-encoding genes, samples were subsampled to 15,000,000 reads per sample.

### 2.6. Statistical Analysis

Stratification and clustering of LPD were carried out using Euclidean distances and general agglomerative hierarchical clustering procedure based on “Ward2”, as implemented in the *gplots* R-package [37]. For univariate data analyses, pairwise comparisons were carried out with unpaired two-tailed Student’s *t*-test, Spearman’s rank coefficient was used for determining correlations, and Chi-Square test for evaluating group distributions. For multivariate data analyses, the association of covariates (e.g., age, BMI, sex) with LPD were assessed by redundancy analysis (RDA) (999 permutations), whereas the association of LPD clusters with GM were analyzed by distance-based RDA (999 permutations) on Canberra distances (implemented in the *vegan* R-package [38] ).

Feature selection for zOTUs was performed with Random Forest. Briefly, for a given training set (training: 70%, test: 30%), the party R-package [39] was run for feature selection using unbiased-trees (cforest_unbiased with 6说000 trees and variable importance with 999 permutations) and subsequently the selected variables were used to predict (6000 trees with 999 permutations) their corresponding test set using *randomForest* R-package [40]. All statistical analyses were performed in R versions ≥3.6.0.

## 3. Results

### 3.1. Participants and Data Collection

Two hundred and sixty-two individuals (men:women 90:172) with ages between 20 and 85 years (Figure 1A) and BMI ranging between 19 and 37 kg/m^2^ (Figure 1B) were included in this study. Subjects are representatives of community dwelling living in the Danish Capital Region. In this study, we included ^1^H NMR spectroscopy based quantified lipoproteins from human blood plasma [6], short-chain fatty acids profiling and GM composition on fecal samples based on 16S rRNA-gene amplicon sequencing and shotgun metagenome sequencing for a subset of samples (Figure 1C).

### 3.2. LPD Profiles, Stratification, and Host Covariates

LPD profiles of the study subjects were predicted from ^1^H NMR measurements of blood plasma. A total of 14 lipoproteins-subfractions, including apolipoprotein A (ApoA1) and apolipoprotein B (ApoB), together with cholesterol, triglycerides (TG), cholesterol ester (CE), free cholesterol and phospholipids, were quantified in VLDL, IDL, HDL, LDL [6]. Linking host covariates and LPD profiles, redundancy analysis (RDA) of LPD profiles showed a significant (*p* ≤ 0.01) effect of age, BMI and sex on LPD profiles (Figure 2B) with a combined size effect of up to 24.6% (Figure 2B,C).

Clustering of LPD profiles segregated study participants into three groups (Figure 2A, Appendix A). Cluster 1A and 1B were characterized by higher concentrations of LDL sub-fractions and their constituents (particularly evident in subclasses 1 and 2). Clusters 1A and 2, on the other hand, were characterized by lower concentrations of HDL sub-fractions (associated with HDL2a and HDL2b), whereas higher concentrations of HDL-3 particles in subjects of cluster 1A were observed (Appendix A). Furthermore, plasma concentrations of CE, phospholipids and CE were higher among cluster 1A and 1B. When comparing the plasma fractions of the study participants to the recommendations of cholesterol classes provided by the National Institute of Health (NIH) [41], for clusters 1A and 1B total cholesterol and LDL levels were above the recommendations, while for clusters 1B and 2 the levels of HDL were below the recommended values. No differences (Fisher-test, *p* = 0.9) in the distribution/frequency of participants (*n* = 8) treated with statins across LPD clusters were observed.

LPD profiles were also found to covariate with host attributes, cluster 2 subjects were significantly younger than clusters 1A and 1B (Figure 2D), and cluster 1B showed the lowest BMI (Figure 2E). These results were also consistent even after correcting for sex effects, given that cluster 1B had a significantly higher proportion of women (Fisher test *p* < 0.01, Figure 2A, Appendix A).

### 3.3. LPD Clusters Are Linked with GM Profiles

The GM of study participants (*n* = 262) was profiled using high-throughput amplicon sequencing the V3-region of the 16S rRNA gene (11,544 zOTUs), as well as shotgun metagenome sequencing of total genomic DNA for a subset of samples (*n* = 58). Gene content and functionality (based on KEGG orthologues-KOs) were predicted based on PICRUSt [20] (for 16S rRNA gene amplicons), as well as through ORF calling and gene prediction of assembled contigs reconstructed from shotgun metagenome data. Validation of PICRUSt against metagenome KO yielded a high correlation coefficient (Pearson *r* = 0.77, Figure 3A) between the gene richness of both datasets. Alpha diversity analyses between LPD clusters revealed no significant (*t-*test *p* > 0.05) differences in phylotypes richness (Figure 3B) or KOs richness (Figure 3C). A significant (Dip-test *p* < 0.001) bimodal distribution of KO richness among the study participants was observed (Figure 3C), but a higher-/lower- gene count was not associated to LPD clusters (Figure 3C) or BMI categories (Figure 3D). Significant differences in composition (beta-diversity) between LPD clusters were observed among phylotypes (Canberra distance, Adonis test *p* < 0.05, R^2^ = 0.62–1%).

### 3.4. LPD Clusters Correspond with GM and KOs Features

After feature selection based on random forest, LPD clusters were partially discriminated (Figure 4A) by 206 selected sequence variants (zOTUs) distributed to over 10 families (Figure 4B). Among these, zOTUs affiliated to *Ruminococcaceae* (75) and *Lachnospiraceae* (58) represented 64%, followed by *Bacteroidaceae* (8), *Bifidobacteriaceae* (7), *Christensenellaceae* (6), *Coriobacteriaceae* (5) and four other sparse bacterial families (47). The cumulative abundance (cumulative sum scaling, CSS) of those families showed differences between LPD clusters, with cluster 1A being associated with a higher abundance of *Lachnospiraceae* and a lower abundance of *Christensenellaceae* members, while cluster 1B was characterized by a larger proportion of *Ruminococcaceae* phylotypes, and cluster 2 showed increased proportion of *Bifidobacteriaceae*, *Bacteroidaceae* and reduced abundance of *Coriobacteriaceae* (Figure 4B,C).

KEGG orthologues predicted through PICRUSt demonstrated very weak discrimination power towards LPD clusters (Figure 4D, Appendix A shows detailed 3rd level KEGG functions), this included 54 KOs affiliated to >9 primary and secondary metabolism processes, as well as signaling and cellular processes (Figure 4E). Despite its documented limitations [42], PICRUSt was still able to reveal a decreasing abundance of functional modules among subjects of cluster 1A and 2 as compared to those of cluster 1B (Figure 4E,F). Analysis on aggregated functions per KOs (2nd level KEGG) showed that cluster 1B was characterized by a higher abundance (*t*-test *p* < 0.05) of functions related to metabolism of amino acids (e.g., Phe, Tyr and Trp biosynthesis), carbohydrates (e.g., pyruvate, propanoate, and butanoate metabolism), lipids (glycerolipids and glycerophospholipids metabolism) and genetic information processing (e.g., transcriptional factors) (Figure 4F).

Correlation analyses of selected zOTUs vs LPD profiles displayed several significant (Spearman FDR *p* ≤ 0.05) associations (Figure 4G, Appendix A). Most *Ruminococcaceae* (74/75 phylotypes, mostly unclassified), a division of *Lachnospiraceae* (13/58 phylotypes, mostly unclassified), *Bacteroidaceae* (e.g., *B. massiliensis*, *B. caccae*), *Christensenellaceae* (unclassified genus) and *Coriobacteriaceae* (unclassified genus) showed positive correlations with HDL subfractions and negative correlations with VLDL and LDL (e.g., LDL3, 4, 5, 6). Contrary to this, most *Lachnospiraceae* (45/58), *Veillonellaceae* (e.g., *V. invisus*) and *Bifidobacteriaceae* (e.g., *Bf. adolescentis*, *Bf. bifidum*) phylotypes correlated negatively with HDL subfractions, and positively with subfractions composed of IDL, LDL and VLDL. For KOs vs LPD (Figure 4H, Appendix A), increasing abundance of functions linked to glycerophospholipids metabolism and amino acids (His, Phe, Tyr and Trp) biosynthesis correlated positively with HDL fractions and negatively with LDL and VLDL. Furthermore, the production of glycosphingolipids, biotin (Vit_B7_) and lipopolysaccharides correlated negatively with small LDL subfractions (e.g., LDL3, 4, 5, 6).

### 3.5. Metagenome Bins and Functions Associated with LPD Clusters

Fifty-eight samples randomly selected were subjected to shotgun metagenome sequencing (Figure 1C) generating on average 5.2 GB per sample. ORF calling on the entire assembled dataset of generated ~1.4 million gene-clusters (90% similarity clusters, here termed “genes”), with 84,560 core genes being present in at least 90% of the metagenome sequenced samples. RDA analysis of the core-gene dataset showed significant (*p* = 0.001) differences between LPD clusters and explaining up to 23.7% of the total variance in gene composition (Figure 5A). Ranking of variables (i.e. top 150) within the 1st and 2nd canonical components of the CAP analyses provided an overview of 35 “known” metabolic genes (>90% identity match to the integrated non-redundant gene catalog with KEGG ortholog entries [35], Figure 5B, Appendix A) linked to >10 2nd level KEGG functions, which resembled the large majority of those predicted by PICRUSt (see Figure 4E,F). A higher abundance of these genes was observed among subjects grouped within Cluster 1B relative to cluster 1A and Cluster 2. To determine the species associated with these genes, gene-sequences were mapped back to 1419 metagenome-assembled genomes (MAGs) (Figure 5C). Sixty MAGs affiliated to *Lachnospiraceae*, *Clostridiales*, *Coriobacteriaceae* and unclassified *Firmicutes* clustered within 19 species and were found to contribute with 27 out of the 35 genes that discriminated LPD clusters (Figure 5D, Appendix A). Most *Lachnospiraceae* and *Clostridiales* MAGs (Figure 5D,F, Appendix A) contributed with peptidoglycan and glycan biosynthesis, thiamine (Vit_B1_) and pantothenate (Vit_B5_) metabolism, starch degradation and butyric acid metabolism (butanol dehydrogenase that may lead to increased concentrations of 1-butanol at the expense of butyrate production, Figure 5E). On the other hand, *Eggerthellaceae* MAGs were linked to biosynthesis of glucosinates, metabolism of propionic acid, biosynthesis of fatty acids, Vit_B6_ metabolism, as well as folate (Vit_B9_) biosynthesis (Figure 5D,F, Appendix A). Subjects belonging to LPD-cluster 1B had a significantly higher cumulative abundance of these MAGs (Figure 5H,I) than those in LPD-clusters 1A and 2, whereas their cumulative abundance had significant positive (Spearman *p* < 0.001) correlations with constituents (e.g., Cholesteryl ester) of larger HDL sub-classes (HDL2a and HDL2b) (Figure 5I).

The concentrations of acetate and propionate in fecal samples had no differences between LPD clusters. However, higher concentrations of butyrate, isobutyrate, 2-methylbutyrate, valerate and isovalerate (ANOVA Tukey’s HSD *p* < 0.05) were observed in cluster 2 (Figure 6A–E). To determine whether microbial activity was linked to the production of such branched-chain fatty-acids, we analyzed the abundance of isobutyrate kinase (Appendix A) and 2-methylbutanoyl-CoA (Figure 6F) dehydrogenase in the metagenomic samples (Figure 6F). For 2-methylbutanoyl-CoA dehydrogenase 86% of the gene-variants were also mapped to those 60 MAGs displayed in Figure 5D (ANOVA Tukey’s HSD *p* < 0.05 for cluster 2 LPD subjects), but none of these had significant matches to isobutyrate kinase. Isobutyrate kinase was found in 86 MAGs (Appendix A) belonging to *Bacteroides*, *Ruminococcaceae*, *Alistipes*, *Desulfovibrionaceae* and *Lachnospiraceae*, and whose cumulative relative abundance varied (Appendix A) substantially between LPD clusters.

## 4. Discussion

It is well established that certain LPD profiles are associated with elevated CVD risk, but relatively little is known on the links between GM and LPD. Building on recently published LPD profiles of 262 adult individuals [6] the present study investigates the correlations between LPD-profiles and GM, and its genetic functional assignments.

Stratification of study participants based on their LPD profiles segregated three LPD clusters (C1A, C1B and C2) that corresponded well with levels of total cholesterol, triglycerides, LDL, HDL and VLDL as those recommended by the NIH [41]. Our study demonstrates that lower levels of total HDL are associated with a decrease in the concentration of large subfractions (e.g., HDL2a and HDL2b), while higher levels of total LDL correspond with increased concentration of large LDL subfractions (e.g., LDL1). Furthermore, our results and similar ones [5,10], confirm the association between LPD profiles and host factors like age, sex and BMI [5,10], which altogether can explain up to 25% of the total variance in LPD. Our findings agree with previous reports indicating an inverse association between larger subfractions of HDL and the incidence of metabolic disorders and BMI [43]. As well the positive correlation between the largest HDL and LDL particles with total HDL and LDL cholesterol [44], respectively. In the latter case, a higher concentration of large LDL subfractions (e.g., LDL1) could directly increase the circulating plasma levels of ApoB [44], which has been proposed as a much more sensitive CVD biomarker when compared to LDL cholesterol, but with still major limitations associated to standard measuring [45].

Increasing evidence supports the role of GM to modulate lipids homeostasis and development of dyslipidemia [17,46,47,48]. It has been previously described that dyslipidemia and overweight phenotype may gather a low gut microbial gene-richness (gene diversity and complexity), which displays a bimodal distribution likely related to a low bacterial diversity [49,50]. In our study, despite the fact that such a bimodal distribution was indeed reproduced, no significant differences in gene-frequencies between distinct LPD profiles (e.g., clusters 1A, 1B and 2 clusters), or between lean and overweight participants were observed. In turn, this indicated that the development of remarkably distinct LPD profiles /phenotypes could be related to ecological/structural differences in GM.

GM compositional analysis (Beta diversity) showed significant differences that discriminated LPD clusters/phenotypes (e.g., Figure 4A), and elucidated GM members capable of synthetizing important metabolites (e.g., SCFA and B vitamins). Among those, *Lachnospiraceae* correlated positively with small LDL particles (e.g., LDL3, LDL4 and LDL5), ILDL and VLDL, while *Ruminococcaceae*, a subgroup of *Lachnospiraceae* phylotypes and other less abundant families showed positive correlations with large particles of HDL (e.g., HDL2a and HDL2b). Our findings are in agreement with a recent large-scale study published by Vojinovic et al. [5] who reported that *Lachnospiraceae* and *Ruminococcaceae* members were related to the HDL/LDL ratios.

It is well-established that high HDL levels are associated with a lower risk of developing CVD and the metabolic syndrome [7,8], and mounting evidence support the hypothesis that the heterogeneity of HDL display associations with the same metabolic conditions [7,51,52] that are like mediated by GM. Indeed, recent findings have shown that GM members induce expression of low-density lipoprotein receptors and ApoE in the hepatocytes, facilitating the clearance of triglyceride-rich lipoprotein remnants, chylomicron remnants, and intermediate-density lipoproteins, from circulation [46]. In line with this, our results suggest a link between dyslipidemia and the metabolic potential of MAGs for synthesizing important bioactive compounds such as vitamin B complex, peptidoglycans, and SCFA metabolism. Pantothenate (Vit_B5_), Vit_B6_ and folate (Vit_B9_) have been inversely associated with low-grade inflammation [53] and mortality risk of CVD in a mechanism that may involve regulation of blood homocysteine concentrations [54], and one-carbon metabolism [55]. SCFA such as butyrate and valerate have been shown to decrease total cholesterol and the expression of mRNA associated with fatty acid synthase and sterol regulatory element binding protein 1c, to enhance mRNA expression of carnitine palmitoyltransferase-1α (CPT-1α) in liver [56,57], as well as to ameliorate arteriosclerosis via ABCA1-mediates cholesterol efflux in macrophages [58]. Biosynthesis of peptidoglycans by some GM members has been associated with incidence of stenotic atherosclerotic plaques and insulin resistance [59,60], but recent studies also indicate that these potent signaling molecules play positive roles for enhancing systemic innate immunity [61] and neurodevelopmental processes [62], relaying on a species-dependent fashion [63].

## 5. Conclusions

Collectively our study provides evidence that GM members (e.g., MAGs) and their genes related to the biosynthesis of bioactive molecules needed to carry out lipid metabolism, e.g., vitamin B complex and S/B-CFA, are more abundant among subjects with healthier LPD profiles (e.g., higher HDL2a, HDL2b, and lower LDL). Furthermore, variations in LPD subfractions correlate with differences in the GM composition, but these are not necessarily associated to a higher or lower microbial diversity. Given the limitations of our cross-sectional stud y to elucidate GM-LPDs dynamics, it is not possible to depict the mechanism by which GM may influence variability in LPD subfractions. Future efforts to unravel the processes of LPD particles assembly and their implications in CVD will require an integrative and longitudinal approach of lifestyle factors (diet–type and quantity), physical activity, host genetics/transcriptomics, deep LPDs profiling, and the GM considerations herein provided.

## Figures and Tables

**Figure 1 microorganisms-10-02156-f001:**
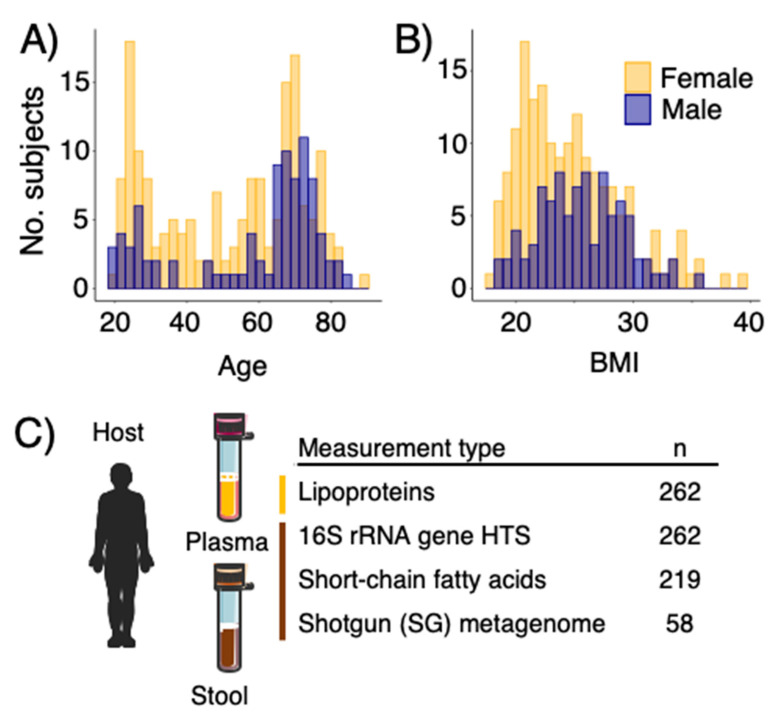
(**A**) Age and (**B**) body mass index (BMI) distribution of the study participants in COUNTERSTRIKE. (**C**) samples and datasets included and analyzed in this study.

**Figure 2 microorganisms-10-02156-f002:**
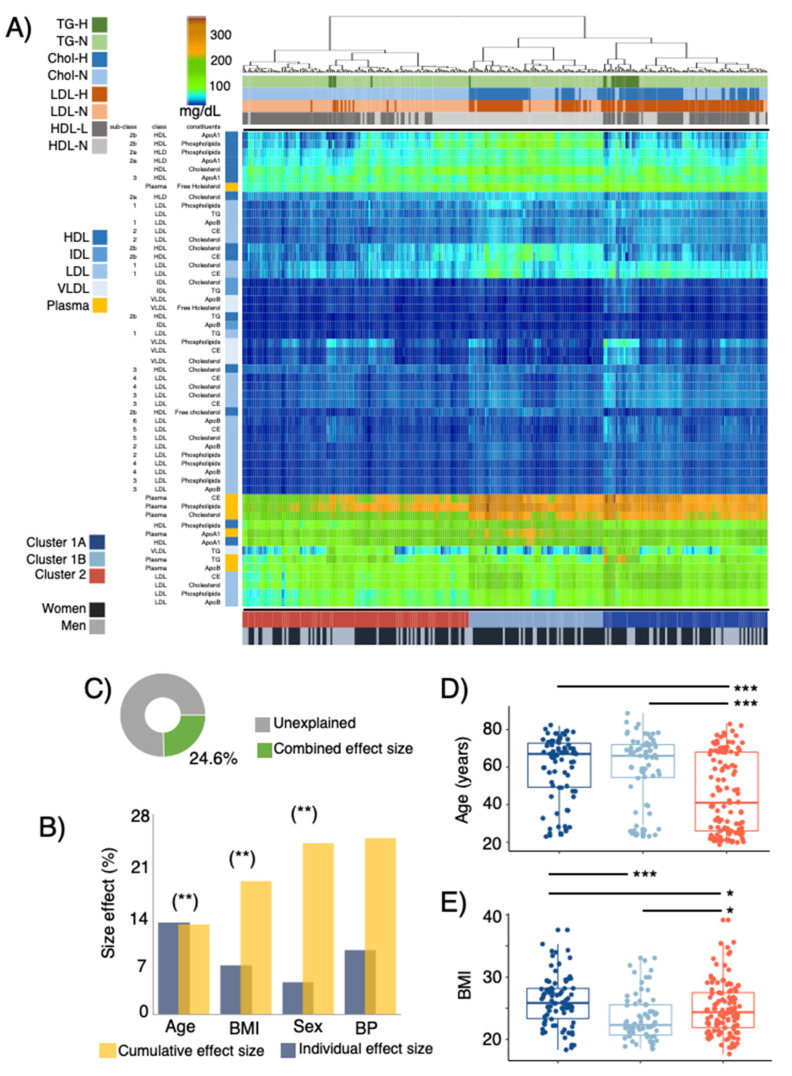
(**A**) Profiles of main and sub-fractions of plasma lipoprotein distribution (LPD) determined by ^1^H-NMR [6]. LPD are clustered using Euclidean distances and general agglomerative hierarchical clustering procedure. Upper color bars represent within-/out- of the recommended levels of main lipoprotein fractions suggested by the NIH [41] (total cholesterol < 200 mg/dL, LDL < 100 mg/dL, HDL > 60 mg/dL, Triglycerides < 150 mg/dL). Lower color bars depict 3 clusters (C1A, C1B and C2) of study participants given their LPD profile and the sex distribution of subjects. (**B**) Cumulative effect size of non-redundant covariates of LPD determined by stepwise RDA analysis (right bars) as compared to individual effect sizes assuming independence (left bars). (**C**) Fraction of LPD variation explained with the stepwise approach. Distribution of (**D**) age and (**E**) body mass index (BMI) between subjects belonging to C1A, C1B and C2. Stars show statistical level of significance (* *p* ≤ 0.05, ** *p* ≤ 0.01, *** *p* ≤ 0.001).

**Figure 3 microorganisms-10-02156-f003:**
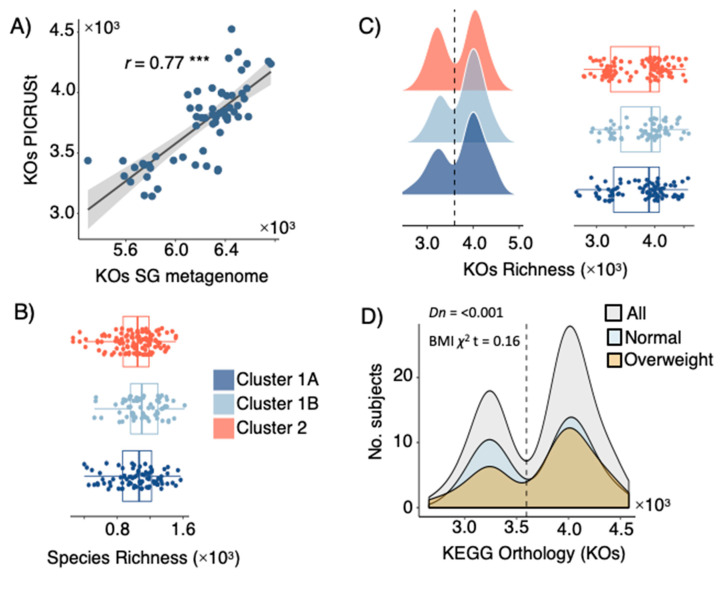
(**A**) Spearman’s rank correlation between fecal microbial KEGG Orthologues (KOs) from shotgun metagenome (SG) sequencing and KO predicted by PICRUSt. (**B**) Richness of microbial phylotypes (zOTUs) richness and (**C**) KO predicted by PICRUSt among subjects catalogued as being C1A, C1B and C2 based on their LPD. (**D**) KO counts (richness) among all subjects and those with BMI ≤ 25 (normal) and BMI > 25 (overweighed); the observed bimodal distribution was statistically significant by the dip-test. Stars show statistical level of significance (*** *p* ≤ 0.001).

**Figure 4 microorganisms-10-02156-f004:**
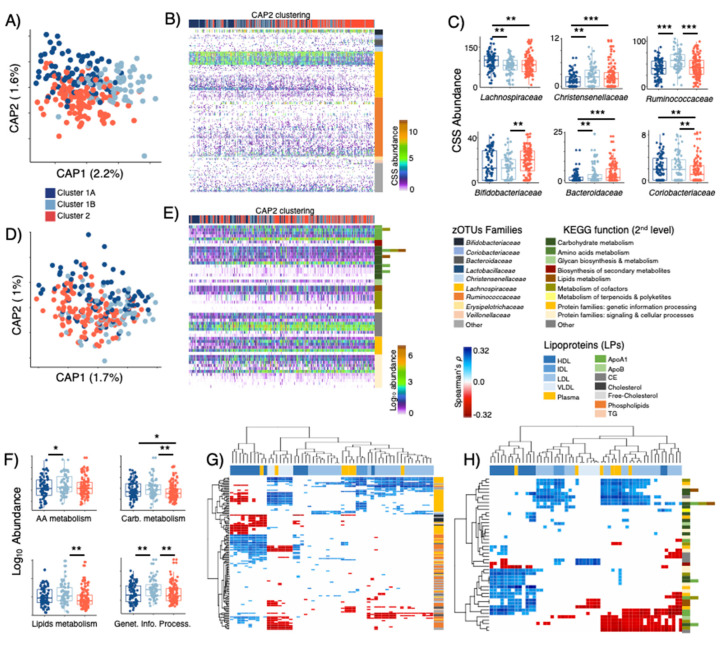
(**A**) Distance-based RDA (Canberra dissimilarity) displaying discrimination of LPD clusters based on selected zOTUs (*p* = 0.001, explained variance = 3.8%) and (**D**) KOs-PICRUSt (*p* = 0.001, explained variance = 2.7%) selected through Random Forests. (**B**) Overview of selected zOTUs and (**E**) KOs-PICRUSt clustered using Canberra distances and general agglomerative hierarchical clustering procedure based on ward2. (**C**) Distribution of zOTUs summarized to family level and (**F**) KOs-PICRUSt summarized to 2nd level KEGG function across subjects belonging C1A, C1B and C2 LPD groups. Heatmaps displaying significant (False Discovery Rate corrected, FDR ≤ 0.05) Spearman’s rank correlations between (**G**) zOTUs and LPD sub-fractions, as well as (**H**) KOs-PICRUSt and LPD sub-fractions. Stars show statistical level of significance (* *p* ≤ 0.05, ** *p* ≤ 0.01, *** *p* ≤ 0.001).

**Figure 5 microorganisms-10-02156-f005:**
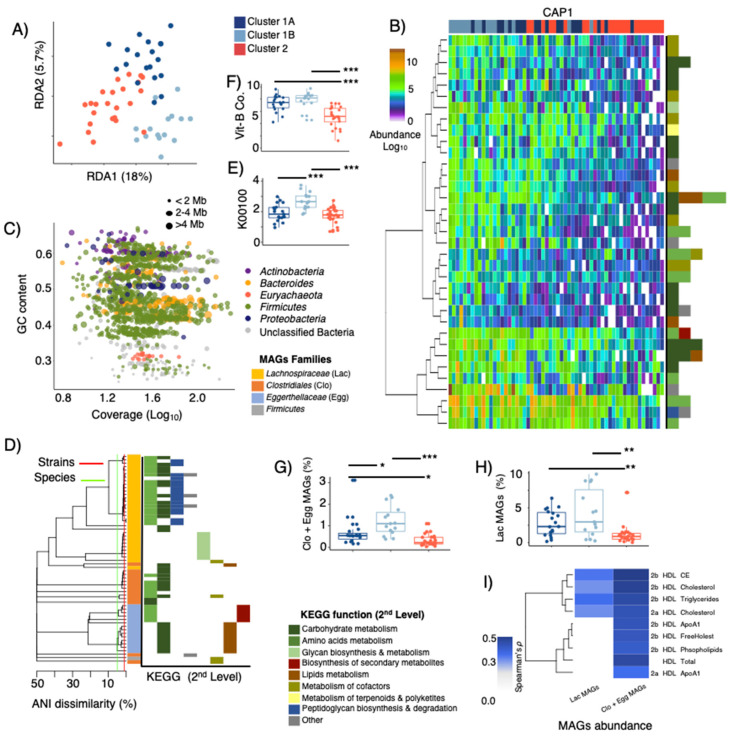
(**A**) RDA displaying discrimination of LPD clusters based on selected KOs obtained from shotgun metagenome and assembly (*p* = 0.001, explained variance = 23.7%). (**B**) Overview of most discriminatory (based on CAP1 and CAP2 within db-RDA) KOs with known metabolic functions clustered using Canberra distances and general agglomerative hierarchical clustering procedure based on ward2. (**C**) GC-content–Coverage plot of metagenome assembled genomes (MAGs) with ≤10% contamination and ≥70% completeness. MAGs are colored according to phylum-level taxonomic affiliation and bubble size indicates their genome size in mega-bases (Mb). (**D**) Phylogeny of MAGs containing KOs that discriminate LPD clusters (1A, 1B and 2), a cut-off of 95-ANI (species-level) and 99-ANI (strain-level) are denoted. MAGs are colored at family level affiliations and their KOs contribution at the 2nd level KEGG function pathways are provided. (**E**) Relative abundance of protein-encoding genes associated with butanol dehydrogenase (K00100), and (**F**) protein-encoding genes associated metabolism and biosynthesis of vitamin B1, B2, B5 and B9. (**G**,**H**) Cumulative abundance (RPKM) of MAGs affiliated to *Clostridiales* (Clo), *Eggerthellaceae* (Egg), and *Lachnospiraceae* (Lac) among LPD clusters. (**I**) Heatmaps displaying significant (False Discovery Rate corrected, FDR ≤ 0.05) Spearman’s rank correlations between MAGs abundance and HDL subfractions. Stars show statistical level of significance (* *p* ≤ 0.05, ** *p* ≤ 0.01, *** *p* ≤ 0.001).

**Figure 6 microorganisms-10-02156-f006:**
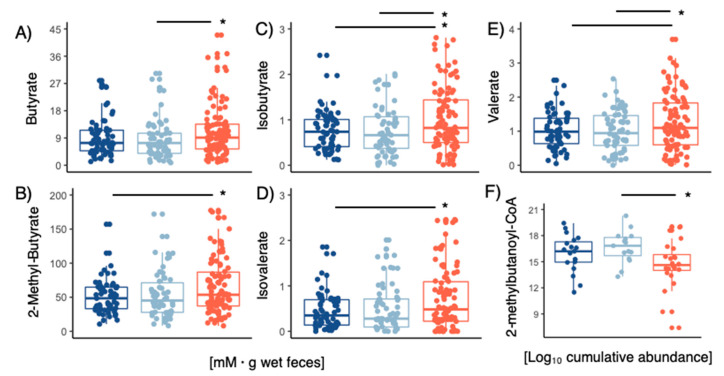
Concentrations of fecal (**A**) butyrate, (**B**) 2-methylbutyrate, (**C**) isobutyrate, (**D**) isovalerate, (**E**) valerate concentrations within the different LPD clusters. (**F**) Cumulative abundance 2-methylbutanoyl-CoA genes screened on metagenomes within LPD clusters. Stars show statistical level of significance (* *p* ≤ 0.05).

## Data Availability

The sequencing dataset generated and analyzed during the current study are available at NCBI under the accession number PRJNA715036. Non-sequencing data generated during and/or analyzed during the current study are available from the corresponding author on reasonable request in accordance with Danish data protection law.

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
