# Peer review of "Gut Microbiome and Its Cofactors Are Linked to Lipoprotein Distribution Profiles"

_microorganisms, 2022, doi:10.3390/microorganisms10112156_

Round 1
Reviewer 1 Report
This study demonstrates that subjects with healthier LPD profiles have higher levels of GM members and their genes related to the biosynthesis of bioactive molecules required for lipid metabolisms, such as vitamin B complex and SCFA. Additionally, variations in LPD subfractions correlate with differences in GM composition.
The paper is very original, well-written, and useful to the scientific community. Congrats to the authors for their intriguing study.
Author Response
Thanks to Reviewer 1 for his/her nice comments.
Reviewer 2 Report
The manuscript “ Gut microbiome and its cofactors are linked to lipoprotein distribution profiles” by Castro-Mejía et al is a very, very interesting report suggesting an association of gut microbiome with lipoprotein distribution profiles; however, there are several issues or misconceptions that need to be corrected or discussed to increase the report´s quality.
A more detailed description of human subjects' characteristics is necessary, they are part of the COUNTERSTRICKE project COUNTERacting Sarcopenia with proTeins and exercise are they under protein or exercise?, what is the CALM cohort? Is there a lipid level inclusion criteria?
How CVD is determined? Is there a lipid profile before inclusion?
Clusters 1A, 1B, and 2 differ in age and lipid levels, (some with levels over the limit) some subjects are overweight or obese. These overweight or obese and dyslipidemic subjects are not healthy subjects were they under treatment (ie. statins)?
Cholesterol and phospholipids, TG are not lipoproteins.
Please explain why only Fifty-eight samples were subjected to shotgun metagenome sequencing and how the samples were chosen
The discussion section is short and, in some manner, limited, as example;
Authors state:
“To the best of our knowledge, this study represents the first to show the contribution of LPD subfractions to the collective levels of cholesterol, cholesterol-types, and triglycerides, as well as recommendations among an age-/BMI- diverse group of apparently healthy adults.” This must be deeply discussed and compared with the values obtained with traditional methodology.
“In relation to alpha-diversity, gene-richness did not show differences among subjects with remarkably distinct LPD profiles (e.g., C1A, C1B and C2 clusters). In addition, despite the fact that a bimodal distribution of gene-richness counts was reproduced as in previous studies [45,46] no significant differences in the gene-frequencies between normal and overweight participants were observed”. This another point to discuss deeply
“Given the cross-sectional nature of our study and its inherent limitations, it is not possible to depict the mechanism by which GM may influence variability in LPD subfractions”. Another point to discuss; is there a difference in food intake (type or quantity)? Are there differences in lipid subfractions between normal and dyslipidemic age and body weight-matched subjects?
Author Response
The manuscript “Gut microbiome and its cofactors are linked to lipoprotein distribution profiles” by Castro-Mejía et al is a very, very interesting report suggesting an association of gut microbiome with lipoprotein distribution profiles; however, there are several issues or misconceptions that need to be corrected or discussed to increase the report´s quality.
A more detailed description of human subjects' characteristics is necessary, they are part of the COUNTERSTRICKE project COUNTERacting Sarcopenia with proTeins and exercise are they under protein or exercise? what is the CALM cohort? Is there a lipid level inclusion criteria?
Valid suggestions. This is now addressed in lines 83-90.
How is CVD determined? Is there a lipid profile before inclusion?
Same as above, this is addressed in lines 83-90.
Clusters 1A, 1B, and 2 differ in age and lipid levels, (some with levels over the limit) some subjects are overweight or obese. These overweight or obese and dyslipidemic subjects are not healthy subjects were they under treatment (ie. statins)?
This is a highly relevant concern. We agree that the “healthy” term may not fit all subjects of the cohort, therefore, we now avoid generalizing the study participants as healthy and describe them instead by their diverse phenotypes (e.g., lines: 19, 31).
In relation to medications/treatments, only 8 individuals (all 65+ years) were treated with statins. However, the distribution/frequency of participants under statin-treatment was not significantly different (e.g., Fisher test/Chi-square test | p-Val > 0.90) across clusters 1A (2 individuals under treatment), 1B (2 individuals under treatment), 2 (4 individuals under treatment). We have added relevant information in the manuscript in lines: 195-196.
Cholesterol and phospholipids, TG are not lipoproteins.
Corresponding changes in lines 178-180 have been made.
Please explain why only Fifty-eight samples were subjected to shotgun metagenome sequencing and how the samples were chosen
The “reduced” number of shotgun metagenomes is related to funding. Samples were randomly selected. This has been added in line 260.
The discussion section is short and, in some manner, limited, as example.
Authors state:
“To the best of our knowledge, this study represents the first to show the contribution of LPD subfractions to the collective levels of cholesterol, cholesterol-types, and triglycerides, as well as recommendations among an age-/BMI- diverse group of apparently healthy adults.” This must be deeply discussed and compared with the values obtained with traditional methodology.
Corresponding changes in lines 404-411 have been made.
“In relation to alpha-diversity, gene-richness did not show differences among subjects with remarkably distinct LPD profiles (e.g., C1A, C1B, and C2 clusters). In addition, despite the fact that a bimodal distribution of gene-richness counts was reproduced as in previous studies [45,46] no significant differences in the gene-frequencies between normal and overweight participants were observed”. This another point to discuss deeply
Corresponding changes in lines 413-423 have been made.
“Given the cross-sectional nature of our study and its inherent limitations, it is not possible to depict the mechanism by which GM may influence variability in LPD subfractions”. Another point to discuss; is there a difference in food intake (type or quantity)? Are there differences in lipid subfractions between normal and dyslipidemic age and body weight-matched subjects?
Corresponding changes in lines 474-480 have been made.
Reviewer 3 Report
The article is interesting. It is well written and organized It worth consideration after revision.
1- The following articles should be cited in the introduction
https://www.ncbi.nlm.nih.gov/pmc/articles/PMC6947520/
https://www.ahajournals.org/doi/10.1161/circresaha.115.306807
2- CVD full name should be stated at the first mentioning , line 80
3- Study limitation and future research plan should be highlighted
4- Conclusion should not contain references number. It can be replaced by as confirmed by the outcomes of the study of [ author name et.al., ] .
5- Date for the ethical approval for the study should be provided.
All my best wishes
Author Response
The article is interesting. It is well written and organized It worth consideration after revision.
1- The following articles should be cited in the introduction
https://www.ncbi.nlm.nih.gov/pmc/articles/PMC6947520/
https://www.ahajournals.org/doi/10.1161/circresaha.115.306807
Thanks for the suggestions. The first article has been cited accordingly (e.g., lines 54-55), while the second article was already cited throughout the manuscript (e.g., lines 51 and 62).
2- CVD full name should be stated at the first mentioning , line 80
This has been addressed.
3- Study limitation and future research plan should be highlighted
This is now highlighted in lines 474-480
4- Conclusion should not contain references number. It can be replaced by as confirmed by the outcomes of the study of [ author name et.al., ] .
Changes have been incorporated accordingly
5- Date for the ethical approval for the study should be provided.
This has been added in line 508
Round 2
Reviewer 2 Report
The manuscript was improved, Authors respond correctly to all my suggestions